Pulmonary anatomy in the Nile crocodile and the evolution of unidirectional airflow in Archosauria

Schachner Emma R. 1 eschachner@gmail.com
Hutchinson John R. 2
Farmer CG 1
1 Department of Biology, University of Utah , Salt Lake City, UT , USA
2 Structure & Motion Laboratory, Department of Comparative Biomedical Sciences, The Royal Veterinary College , Hatfield, Hertfordshire , United Kingdom
Farke Andrew
Electronic publication date: 2013 Mar 26
Publication date: 2013
Volume: 1
Electronic Location ID: e60
Received 2012 Nov 13; Accepted 2013 Mar 10
Copyright: © 2013 Schachner et al.
Copyright year: 2013
Copyright holder: Schachner et al.
License: This is an open access article distributed under the terms of the Creative Commons Attribution License, which permits unrestricted use, distribution, and reproduction in any medium, provided the original author and source are credited.
License URL: https://creativecommons.org/licenses/by/3.0/

Keywords: Lung, Respiration, Anatomy, Evolution, Pneumaticity, Endothermy, Crocodylia, Bronchi

Funding: American Association of Anatomists Postdoctoral Fellowship American Philosophical Society Franklin Research Grant National Science Foundation grants IOS-1055080 and IOS-0818973 This work was supported by an American Association of Anatomists Postdoctoral Fellowship and an American Philosophical Society Franklin Research Grant to ERS, and National Science Foundation grants to CGF (IOS-1055080 and IOS-0818973). This work was also supported by a generous gift from Sharon R. Meyer. The funders had no role in study design, data collection and analysis, decision to publish, or preparation of the manuscript.

==============================
The lungs of birds have long been known to move air in only one direction during both inspiration and expiration through most of the tubular gas-exchanging bronchi (parabronchi). Recently a similar pattern of airflow has been observed in American alligators, a sister taxon to birds. The pattern of flow appears to be due to the arrangement of the primary and secondary bronchi, which, via their branching angles, generate inspiratory and expiratory aerodynamic valves. Both the anatomical similarity of the avian and alligator lung and the similarity in the patterns of airflow raise the possibility that these features are plesiomorphic for Archosauria and therefore did not evolve in response to selection for flapping flight or an endothermic metabolism, as has been generally assumed. To further test the hypothesis that unidirectional airflow is ancestral for Archosauria, we measured airflow in the lungs of the Nile crocodile (Crocodylus niloticus). As in birds and alligators, air flows cranially to caudally in the cervical ventral bronchus, and caudally to cranially in the dorsobronchi in the lungs of Nile crocodiles. We also visualized the gross anatomy of the primary, secondary and tertiary pulmonary bronchi of C. niloticus using computed tomography (CT) and microCT. The cervical ventral bronchus, cranial dorsobronchi and cranial medial bronchi display similar characteristics to their proposed homologues in the alligator, while there is considerable variation in the tertiary and caudal group bronchi. Our data indicate that the aspects of the crocodilian bronchial tree that maintain the aerodynamic valves and thus generate unidirectional airflow, are ancestral for Archosauria.

The evolution of the clade Archosauria is a popular subject of scientific study because of the dramatic evolutionary radiations that characterize it. Archosauria includes many extinct lineages, such as crocodyliforms and other pseudosuchians (ornithosuchids, aetosaurs, and poposaurs), pterosaurs, and non-avian dinosaurs. The latter grade came to dominate numerous niches on land during the Mesozoic Era, but were supplanted by mammals after the Cretaceous-Paleogene extinctions. Whereas birds survived the Mesozoic to become extremely widespread, multiple lineages of crocodyliforms persisted at much lower levels of diversity yet a nonetheless impressive global distribution. Explanations for the archosaur radiation are complicated by a puzzling pattern of faunal turnover. Did archosaurs evolve particular features that gave them a competitive advantage that enabled them to supplant the synapsids, or did they flourish opportunistically in the wake of the massive Permo-Triassic extinctions? Many aspects of the archosaurian radiation, such as morphological variety, rates of morphological character evolution, faunal abundance, and taxonomic diversity, have been examined in an increasingly rigorous manner, providing insight into the patterns and processes underpinning this radiation (Nesbitt, 2011). However, these patterns are consistent both with changes that are predicted when a lineage is presented with new ecospace and when it evolves an innovative trait (Brusatte et al., 2010).

The respiratory system is the interface between the environment and the internal milieu and is the first step in the oxygen cascade. It is thought to be particularly derived in birds because it consists of air sacs that function as bellows and lungs composed of a series of open-ended tubes (dorso, ventro, and parabronchi) (Duncker, 1971). Furthermore, airflow through most of these tubes occurs in one direction during both phases of the respiratory cycle due to the presence of aerodynamic valves (Butler, Banzett & Fredberg, 1988; Wang et al., 1988). Conventional wisdom has attributed the evolution of these features of the avian respiratory system to the high energetic demands of flight (Maina, 2000). Alternatively, it is possible that endothermy, rather than flight per se, underpins the evolution of these features. Various aspects of the avian respiratory system have been proposed to have evolved in the ornithodiran lineage before the evolution of birds (Huxley, 1882; Perry, 1992; Bonde & Christiansen, 2003; O’Connor & Claessens, 2005; Wedel, 2007; Sereno et al., 2008; Claessens, O’Connor & Unwin, 2009; O’Connor, 2009; Wedel, 2009; Butler, Barrett & Gower, 2012; Yates, Wedel & Bonnan, 2012). For example, in discussing the dorsal expansion of the lungs in extant archosaurs in 1882, Huxley proposed that, “It seems not improbable that the great height of the bodies and arches of the anterior thoracic vertebrae in some Dinosaurians may be connected with a similar modification of the lungs.” The recent discovery of unidirectional airflow in the lungs of alligators (Farmer, 2010; Farmer & Sanders, 2010) suggests the character of unidirectional airflow through open-ended, tubular gas-exchanging structures is older than the ornithodiran lineage, and predates the evolution of avian style air sacs, having evolved in the common ancestor of the pseudosuchian and ornithodiran lineages. Unidirectional airflow and the structures requisite for aerodynamic valves have been proposed to have arisen in the ectothermic ancestors of these lineages and to have functioned as a means to couple the motion of the beating heart with airflow during periods of breath-holding (apnea) (Farmer, 2010). In this scenario, the unidirectional airflow found in birds, which appears to facilitate their ability to fly in hypoxic conditions (Meyer, Scheid & Piiper, 1981), is an exaptation, having originally served their distant ancestors in a completely different role, that of facilitating gas exchange during apnea.

Levels of atmospheric oxygen have probably played a large role in the evolution of life throughout the Phanerozoic (e.g., Graham et al., 1995; Huey & Ward, 2005; Berner, VandenBrooks & Ward, 2007; Kaiser et al., 2007). The archosaur lung may be a key innovation that gave archosaurs a competitive advantage over the synapsids in niches that required highly aerobic metabolisms during the atmospheric hypoxia of the Triassic (Farmer, 2010; Farmer & Sanders, 2010). The avian lung has long been thought to be an adaptation for the high aerobic demands of flapping flight (Maina, 2000). Furthermore, features of the avian lung, such as unidirectional airflow, appear to improve the efficacy of gas exchange under conditions of hypoxia (Meyer, Scheid & Piiper, 1981). If pulmonary aerodynamic valves and unidirectional airflow were present in basal archosaurs, these characters could have given the entire lineage a selective advantage under conditions of hypoxia. Thus key innovations in the respiratory system may have enabled the archosaurs to usurp the synapsid dominance in ecomorphological niches that required high aerobic capacities (Farmer, 2010).

A considerable amount of work has been done on the macro- and microscopic anatomy of reptilian lungs (Broman, 1939; Duncker, 1978; Perry & Duncker, 1978; Klemm et al., 1979; Perry, 1998), including that of crocodilians (Lereboullet, 1838; Cuvier, 1840; Milani, 1897; Perry, 1988; Perry, 1990; Perry, 1998). However, less attention has been focused on the anatomy of the crocodilian bronchial tree (Sanders & Farmer, 2012) despite long known similarities to avian primary and secondary bronchi (Huxley, 1882; Moser, 1902; Broman, 1939; Boelert, 1942; Perry, 1988; Perry, 1989; Perry, 1992; Perry, 2001; Farmer, 2010; Farmer & Sanders, 2010; Sanders & Farmer, 2012). For example, in describing the crocodile lung Huxley (1882) states,

“Each bronchus is continued directly backwards into a wide canal, which dilates into an oval sac-like cavity at the posterior end of the lung, representing the mesobronchium with the posterior air-sac in birds. In the dorsal and mesial wall of the mesobronchium there are five or six apertures, which lead into as many canals, representing the entobronchia (ventrobronchi) in birds. These pass, the anterior two almost directly forwards, and the others more or less obliquely, to the dorsal margin; and they lie quite superficially on the mesial face of the lung. The first is very much larger than the others, and ends in a dilatation at the anterior end of the lung. It is united with the second by transverse branches. Along the ventral margin of the lung there are four sac-like chambers, which communicate, in the case of the two anterior, with the entobronchia, and, in the case of the two posterior, with the mesobronchium. Finally, there are two very large canals, external to these, which communicate with the mesobronchium by large apertures in its dorsal wall, and give off branches to the outer face of the lung, representing the ectobronchial (dorsobronchial) system of birds. The orifices with which the surfaces of all these canals, except the anterior half of the mesobronchium, are thickly set, lead into depressions, which are often so deep as to become cylindrical passages, simulating the parabronchia of birds. Thus, notwithstanding all the points of difference, there is a fundamental resemblance between the respiratory organs of Birds and those of Crocodiles, pointing to some common form (doubtless exemplified by some of the extinct Dinosauria), of which both are modifications.”

Based on anatomical data, Perry (1988) suggested that the airways in the Nile crocodile end blindly forming chambers rather than open ended tubes, and therefore airflow must be tidal. To date three studies have measured airflow patterns in the American alligator (Alligator mississippiensis, clade Alligatoridae) (Bickler et al., 1985; Farmer, 2010; Farmer & Sanders, 2010). Using scintigraphy, Bickler and colleagues (1985) described a radial spread of gas from the intrapulmonary bronchus into a multicameral alligator lung, and tidal airflow. In contrast, direct measurements of airflow in A. mississippiensis demonstrated that gases move unidirectionally through most of the secondary bronchi (Farmer & Sanders, 2010). These data indicated that the previous understanding of the relationship between anatomical architecture and airflow patterns in the lung of Alligator was incorrect. The lung is not composed of multiple chambers (multicameral) that end blindly, but of open ended tubes. Furthermore, the presence of unidirectional airflow in crocodilians suggests that this pattern of airflow is basal for the entire clade Archosauria.

To gain insight into basal archosaur pulmonary anatomy, and to elucidate how and why the lungs of birds and those of the American alligator diverged, requires the careful study of a range of crocodilian and avian species. Whereas numerous studies are available for both anatomical and physiological aspects of avian lungs (Duncker, 1971; Brackenbury, 1972; Maina & Nathaniel, 2001; Maina, 2006; Farmer & Sanders, 2010), there are few studies of the crocodilian respiratory system, particularly studies that combine physiological and anatomical measurements. The clade Crocodylia is composed of at least two major lineages: Alligatoroidea, which includes the two extant alligator species and seven extant caiman species and Crocodyloidea, which includes the 13+ extant species of crocodiles. However, the phylogenetic position of a potential third lineage, the Gavialidae (gharials), remains controversial (lying outside Alligatoroidea + Crocodyloidea, or within the latter clade (Brochu, 1997; Gatesy et al., 2003; Oaks, 2011)). Identification of key features that are common to all the crocodilian lineages and those that vary interspecifically necessitates detailed study of species from each lineage. Here, we report the results of detailed study on the anatomy and airflow patterns in the lungs of the Nile crocodile (Crocodylus niloticus), the first such analysis of a non-alligatoroid crocodylian.

Materials and Methods

We collected data from seven specimens of Crocodylus niloticus and five specimens of Alligator mississippiensis for comparison. Approval for this study was granted from the University of Utah Institutional Animal Care and Use Committee (IACUC), protocol number 10-12003. C. niloticus were obtained post mortem (varied, natural causes but no respiratory pathology) from the conservation and breeding center “La ferme aux crocodiles” (Pierrelatte, France), with specimen identifiers FNC6 (10.1 kg), NNC1 (3.2 kg), NNC3 (1.01 kg), NNC4 (14.6 kg), NNC5 (0.5 kg), NNC6 (0.8 kg), and NNC9 (0.58 kg). The five alligators were obtained from the Rockefeller Wildlife Refuge in Louisiana: 2.3 kg, 3.6 kg, 5.4 kg, 3.6 kg, 11 kg, and 64 inches long (mass unknown). The C. niloticus lungs were excised and soaked in an iodine potassium iodide (I2KI) solution at concentrations varying from 2.25–3.75% (Jeffery et al., 2011) prior to the CT scans. NNC5 and NNC6 were inflated and scanned in a medical grade CT unit at the Royal Veterinary College, London at 90 kVp and 133MA with a slice thickness of 0.75 mm; NNC9 was scanned in a micro CT unit at the University of Cambridge with a slice width of 0.0816 mm. We imaged live unsedated adult alligators during a natural apnea phase at the University of Utah Medical Center using a 164 slice dual energy Siemens SOMATOM Definition computed tomography unit. Image acquisition parameters: slice thickness = 0.6 mm and 1 mm, kVp 120, MA 200. The pulmonary bronchi were segmented into a 3D model from DICOM image files with the visualization software Avizo 7.

To measure flow, dual heated thermistor airflow probes were individually implanted in each of the secondary bronchi (Fig. 1) after the anatomy was mapped out using computed tomography and Avizo 7.0 software. The probe was connected to an air flow meter (HEC 132C Thermistor Flowmeter, Hector Engineering Co., Inc., Elletsville, IN), and the signal transformed from analog to digital (Biopac Systems Inc, Goleta, CA), and then recorded on a MacIntosh G4 Powerbook laptop using AcqKnowledge software (Biopac Systems Inc, Goleta, CA). Airflow in and out of the trachea was measured with a pneumotach (Hans Rudolph Inc., Shawnee, KS). Measuring airflow in excised lungs in crocodilians has been validated and produces the same results as in vivo experiments (Farmer, 2010; Farmer & Sanders, 2010); thus only ex vivo lungs were used here. Once the probe was in place (Fig. 1), air was pushed into the excised lung using a syringe to measure airflow in and out of the primary and secondary bronchi.

Figure 1 Excised lungs from a 3.1 kg Nile crocodile (NNC1; Crocodylus niloticus) demonstrating probe placement, head is to the right. (A) Lungs in ventral view with the probe inserted in the right ventrobronchus (CVB); (B) the lungs in right dorsolateral view with the probe inserted in the second dorsobronchus (D2). Scale bars = 1 cm.

Results

Trachea and carina

The trachea contains cartilaginous rings, and is very distensible along its long axis (Fig. 2A). It is centered approximately ventral to the esophagus in smaller individuals (≤0.5 kg) but lies to one side in the larger animals (5–15 kg). In the smaller specimens, the carina is located just cranial to the outlet of the great vessels from the pericardium. In the larger animals, it is positioned much more proximally and due to an elongation of the primary bronchi, the trachea forms a distinct loop (Fig. 2B).

Figure 2 (A) Inflated lungs of a 1.01 kg Crocodylus niloticus (NNC3) in ventral view, head is to the right; (B) ventral view of the tracheal loop and heart (pericardium has been removed) of a 10.1 kg C. niloticus (FNC6), head is to the right. Scale bars = 1 cm. Arrows indicate tracheal loop; or lack thereof in the smaller individual (A).

Primary bronchi

The primary bronchi are composed of three distinct parts: the extrapulmonary primary bronchi, the cartilaginous intrapulmonary primary bronchi and the non-cartilaginous intrapulmonary primary bronchi. The extrapulmonary primary bronchus enters the lung ventro-medially at approximately one third the length of the lung from its apex, and courses in a drawn out S-shaped curve laterally, caudally, and dorsally. In all of the specimens examined, the non-cartilaginous portion of the intrapulmonary primary bronchus broadens significantly to become at least twice as wide as the cartilaginous region as it extends caudally; it then loops medially at the caudal end of the lung generating a distinctive hook-like bronchus. At the caudal margin of the hook in all specimens, the primary bronchi balloon out caudally into sub-equal caudally positioned sac-like structures, in both lungs (Fig. 3). The caudal region of the lung in Crocodylus niloticus is less vascularized than the dorsal regions and as a result is likely less involved in gas exchange (Perry, 1990).

Figure 3 3D segmented surface models of the bronchial trees of Crocodylus niloticus demonstrating the position of the caudal expansion of the caudal saccular regions of the primary bronchi within the lung, all in dorsal view. (A) The translucent lung surface and bronchial tree of NNC9; (B) the bronchial tree of NNC9; (C) the bronchial tree of NNC5; (D) the bronchial tree of NNC6. Abbreviations: CVB, cervical ventral bronchus; CSS, caudal sac-like structure; D2-D7, dorsobronchus 2-7; Ls, lung surface; Pb, primary bronchus. Bronchial trees are not to scale relative to one another.

Secondary bronchi

There are several types of secondary bronchi (Fig. 4). They differ due to the location within the lung and by their airflow patterns.

Figure 4 Segmented airways and lung surface of a 0.5 kg specimen of Crocodylus niloticus (NNC9) generated from a µCT scan in left craniolateral view. The solid airways are visual representations of the negative spaces within the lung. (A) The primary, secondary, and tertiary bronchi positioned with respect to the lung surface (transparent blue); (B) the primary, secondary and tertiary bronchi; (C) the primary and secondary bronchi. For a detailed model of the anatomy see Figs. 5 and 6. Color scheme: translucent blue, lung surface; white, trachea and primary bronchi; mint green, cervical ventral bronchi (CVB); lime, D2; neon green, D3; aqua, D4; light aqua, D5; light blue, D6, periwinkle, D7; blue, laterobronchi; purple, caudal group bronchi (CGB); red, M1; neon pink, M2; medium pink, M3; light pink, M4; pale pink, M5; pale purple-deep pink-purples, M6-8; yellow-gold, cardiac lobes.

Cervical ventral bronchi (CVB; D1)

The most proximal and first ostium on the primary bronchus is very close to the hilus and opens on a largely lateral location on the primary bronchus into a conical vestibule. This cone makes a hairpin turn into a cranially directed and large diameter bronchus. This bronchus is the ventrobronchus (the CVB), or D1 (the D1 is from Broman’s (Broman, 1939) identification as the first dorsal branch off of the primary bronchus) (Figs. 5A–5D). The CVB arches cranially so that the main body of the bronchus lies almost parallel to the trachea. There is some variability in the overall morphology of the CVB from individual to individual and even between the right to left lungs. In some individuals (e.g., NNC9; Figs. 5A–5D and 6A–6D), there is a large hook on the distal tip of the CVB that arches dorsally then caudally towards the distal tip of D2.

Figure 5 Primary bronchi, ventrobronchi (CVB), dorsobronchi (D), and medial bronchi (M) of a 0.5 kg Crocodylus niloticus (NNC9) generated from µCT. The ventrobronchus and dorsobronchi in (A) left craniolateral view; and (B) left lateral view. The ventrobronchus and medial bronchi in (C) right craniolateral view; and (D) left lateral view. The solid airways are visual representations of the negative spaces within the lung. Abbreviations: CVB, cervical ventral bronchus; D2-7, dorsobronchi 2-7; M1-8, medial bronchi 1-8; Pb, primary bronchus; R, right; Tr, trachea.

Dorsobronchi (D2-X)

The dorsobronchi arise sequentially via large oval-shaped openings (termed macroostia (Sanders & Farmer, 2012)) from the dorsal and dorsolateral surface of the cartilaginous intrapulmonary primary bronchi and variably up to one half of the proximal part of the non-cartilaginous intrapulmonary primary bronchi. Along with the CVB, they are the largest bronchi in the lung, arching dorsally and then cranially (Figs. 5A and 5B). Crocodylus niloticus has between four and six dorsobronchi; however, there is individual variation, as well as bilateral variation between the right and left sides with regard to both number and specific bronchial morphology. In all specimens, D2-D4 are long tubular bronchi with a wide base that arch dorsally and then run cranially towards the apex of the lung. The more caudal dorsobronchi (D5-7) run dorsally or dorsolaterally from their origin and are generally half the length (longitudinally) of the proceeding three. They also often exhibit more branching, intermediate between D2-4 and the laterobronchi in one specimen (NNC9).

M bronchi (M1-X)

The M, or medial bronchi exhibit a similar morphological pattern to that of the dorsobronchi, but have a medial origin from the cartilaginous intrapulmonary primary bronchi. There is more bilateral asymmetry in M bronchi between the right and left lungs in Crocodylus niloticus, with variation in both the number of branches (six to eight) and overall branch morphology (Figs. 5C and 5D). In all three specimens, M1 is the second branch off of the primary bronchus. It maintains a long, tubular anatomy and runs dorsocranially in unison with D2. The subsequent branches vary from individual to individual, but follow an overall trend: the more cranial branches are tubular and have a wide, rounded base; the middle bronchi pass dorsally giving off both cranial and caudal forks; the caudal M bronchi arch caudally and then caudoventrally terminating in a sac-like tip.

Laterobronchi

Multiple small ostia along the ventral and lateral surface of the intrapulmonary primary bronchi open up into sac-like secondary bronchi (Figs. 6A–6D). These laterobronchi have very small, constricted openings that balloon out into large chambers containing multiple finger-like protrusions that extend in all directions. The laterobronchi vary in size, number and morphology between the right and left lungs as well as among individual specimens.

Figure 6 The primary bronchi, ventrobronchi, cardiac lobes, laterobronchi, and caudal group bronchi of a 0.5 kg Crocodylus niloticus (NNC9) generated from µCT. The lungs in (A) left craniolateral view; (B) dorsal view; (C) left lateral view; (D) ventral view. The solid airways are visual representations of the negative spaces within the lung. Abbreviations: C1-4, cardiac lobes 1-4; CGB, caudal group bronchi; CVB, cervicoventrobronchi; L, laterobronchi; Tr, trachea.

Caudal Group Bronchi (CGB)

The number and morphology of the CGB are very variable across the individual animals examined; however, there were a few relevant invariable characters. The CGB are the most numerous type of secondary bronchus, maintain a tube-like morphology, and branch in all directions from the non-cartilaginous primary bronchi (Figs. 6A–6D). In Crocodylus niloticus, the CGB extend caudally from the hook of the primary bronchus to the caudal margin of the lung. Like the alligator, these bronchi are significantly less vascularized than the dorsobronchi and cranial M bronchi.

Large diameter tertiary bronchi

The CVB and D2-4 all give off major tertiary branches, the majority arising from the base and proximal third of all five secondary bronchi (Fig. 4B). The largest of these tertiary branches run cranioventrally from all four secondary bronchi to the ventral surface of the lung where they then balloon out much like the laterobronchi. These tertiary bronchi are non-contiguous with the laterobronchi but generate a sequence of chamber-like air sacs that occupy the mid-to caudoventral region of the lungs. Smaller more tubular tertiary bronchi emerge from all of the dorsobronchi and M bronchi along their entire length. In both lungs, tertiary bronchi branch off of the M bronchi to contribute to the cardiac lobes (=pericardiac air sacs). There are (variably) three to four bronchi that contribute to the left and right cardiac lobes in Crocodylus niloticus, which adhere to the dorsal surface of the pericardium.

Small diameter anastomosing bronchi (parabronchi)

The parabronchi are small tubular bronchi that interconnect the secondary bronchi forming a loop between the dorsobronchi and the CVB (Figs. 7B and 7C). These small parabronchi also variably anastamose with adjacent large secondary bronchi. There appears to be a diastema between the origination of the CVB and first dorsobronchus (D2) and the emergence of the first parabronchus interconnecting the two bronchi.

Figure 7 Lungs of a 0.5 kg specimen of Crocodylus niloticus (NNC9) injected with white latex, demonstrating the parabronchi (p) connecting the CVB and D2. (A) Lateral view of the right lung; (B) medial view of the sagittally-sectioned right lung stretched to expose the parabronchi indicated by the pink lines; (C) medial view of the sagittally-sectioned left lung. Pink arrows indicate the parabronchi. Scale bar in A and B = 1 cm; scale bar in C = 1.8 mm. Abbreviations: CVB, cervical ventral bronchus; D2-3, dorsobronchi 2-3; L, laterobronchi; P, parabronchi.

Airflow patterns in the major secondary bronchi

Airflow was measured in four of the large secondary bronchi in five individual specimens of Crocodylus niloticus. In the three dorsobronchi that arise sequentially along the primary bronchi caudal to the CVB (D2-4), air travels caudally to cranially during both phases of the respiratory cycle (Figs. 8A–8E and 9). In the CVB, the first bronchus to arise off of the primary bronchus, air flows cranially to caudally during both phases of respiration in all specimens (Figs. 8G, 8H and 9) (inhalation and exhalation). The dorsobronchi connect to the CVB via the parabronchi (Fig. 7), generating the continuous loop that maintains this airflow pattern (Fig. 10).

Figure 8 Airflow in the dorsobronchi and ventrobronchi measured in excised lungs with dual thermistor flow meters. A positive trace indicates that flow is caudal to cranial (black arrow); a negative trace shows airflow that is cranial to caudal (white arrow). (A) Direction of flow in D2 from NNC6; (B) direction of flow at the trachea while flow was recorded in D2 in NNC6; (C) direction of flow in D3 from NNC6; (D) direction of flow at the trachea while flow was recorded in D3 in NNC6; (E) direction of flow in D4 from NNC5; (F) direction of flow at the trachea while flow was recorded in NNC5 (G) direction of flow at the trachea while flow was recorded in the CVB in NNC5; (H) direction of flow at the trachea while flow was recorded in the CVB in NNC5.

Figure 9 3D segmented models of the bronchial tree of a 0.6 kg specimen of Crocodylus niloticus (NNC6) demonstrating the direction of airflow in the ventrobronchi and dorsobronchi in which airflow has been directly measured during both inspiration and expiration. (A) The primary, secondary, and tertiary bronchi in left lateral view; the color scheme is as in Figs. 2, 6 and 7. (B) The bronchial tree in left lateral view with the left ventrobronchus (CVB) and first three dorsobronchi highlighted to show direction of airflow. (C) The bronchial tree in dorsal view with the ventrobronchi and first three dorsobronchi highlighted to show direction of airflow. (D) The bronchial tree in dorsal view, with all of the secondary and tertiary bronchi removed except for the secondary bronchi in which airflow was directly measured (CVB, D2-D4). (E) The bronchial tree in left craniolateral view with all of the secondary and tertiary bronchi removed except for the secondary bronchi in which airflow was directly measured (CVB, D2-D4). Color scheme for B–E: blue, airflow is cranial to caudal during both phases of ventilation; green, airflow is caudal to cranial during both phases of ventilation; grey, primary bronchus.

Figure 10 Diagrammatic and highly simplified representation of airflow through the dorsobronchi and ventrobronchi during inspiration (A) and expiration (B) in the crocodilian lung, and inspiration (A) and expiration (D) in the avian lung. The avian model is a modification of the Hazelhoff loop (Hazelhoff, 1951). Arrows denote direction of airflow, white arrows show air flowing through the parabronchi, blue arrows show air entering the trachea, the red circled “X” demonstrates the location of the aerodynamic inspiratory valve (i.e., air does not flow through this location during inspiration). Colors represent hypothesized homologous regions of the lung in both groups. Abbreviations: d, dorsobronchi; P, parabronchi; Pb, primary bronchus; v, ventrobronchi.

Discussion

Gross anatomy

A broad range of terminology has been used for the different pulmonary structures in the lungs of reptiles (Broman, 1939; Perry, 1998; Sanders & Farmer, 2012). The trachea and extrapulmonary primary bronchi are nearly universal terms, but considerable variation exists in the terminology used to describe the intrapulmonary bronchus (mesobronchium by Huxley (1882)) and the second and third generations of branching of the avian lung. Based upon hypotheses of homology between alligators and birds proposed by Sanders & Farmer (2012), observed morphological and functional similarities between Crocodylus niloticus, the American alligator and birds, we have chosen to adopt their nomenclature with a few adjustments that incorporate the original developmental terms that Broman (1939) gave to the secondary bronchi in the alligator. Huxley (1882) and others have tended to name the secondary bronchi of birds according to the topological regions they come to occupy. Thus Huxley described entobronchia that come to occupy the ventral and medial portions of the lung. These were termed “bronches diaphragmatique” of Sappey (1847) because of their association with the avian diaphragm and are referred to as “ventrobronchi” by Duncker (1971) because they occupy the ventral lung regions. The ventrobronchi have their origin in openings in the proximal and dorsal part of the intrapulmonary bronchus. The first ventrobronchus curves sharply around the entrance of the intrapulmonary bronchus and courses cranially to occupy the cranioventral portion of the lung and to communicate with the cranial set of air sacs. The other ventrobronchi also come to occupy caudal and mesial regions of the lung. In contrast the “ectobronchi” of Huxely, “bronches costales” of Sappey and “dorsobronchi” of Duncker are six or seven in number and run laterally and dorsally toward the lateral or costal face of the lung. A third group of bronchi that come to occupy the caudolateral portions of the lung were termed laterobronchi by later authors (e.g. Duncker, 1971; Duncker, 1972). Their ostia arise at the same level of the intrapulmonary bronchus as the dorsobronchial ostia. Arising from walls of the ecto- and entobronchi are round apertures that lead into canals that course more or less at right angles to the surfaces of the bronchi. These canals sometimes anastomose with each other and were termed parabronchi (“canaux tertiaries”, Cuvier, 1840). A further set of tubes arising from the parabronchi were termed intercapillary air-passages (air capillaries, Duncker, 1971).

Much of the gross anatomy of our specimens is consistent with previous work on Nile crocodiles (Perry, 1988; Mushonga & Horowitz, 1996) but with several significant exceptions. As in previous descriptions, the lungs, heart and great vessels, and esophagus occupy the cranial half of the body cavity. Dorsally and cranioventrally, the lung is rounded (Figs. 1B, 2A and 3A); caudally and caudoventrally it has large flat surfaces where it attaches to the pericardium as well as to connective tissue that envelop the liver (Fig. 2B). The lungs are bordered dorsally by the vertebral column, ventrally by the sternum and sternal ribs, medially by the mediastinum, and laterally by the dorsal ribs. Perry (1988) reports that in the Nile crocodile, the right and left lungs are mirror images of each other reflected in the sagittal plane, however our observations of this taxon differ in that the right lung was noticeably larger than the left. Mushonga & Horowitz (1996) examined 22 specimens of Crocodylus niloticus and also report that the right lung was larger and longer than the left. Asymmetry is also seen in the position of the trachea: in all of the larger specimens, it runs down one side of the esophagus, making hairpin turns of varying degrees before entering the lungs, whereas in the smaller individuals the trachea lacks this loop and courses along the ventral midline. These observations are consistent with reports of Reese (1915), who states that in many Crocodylia (e.g., Crocodylus vulgaris) the trachea forms a loop before hatching, whereas in other species the loop forms long after hatching. The trachea, extrapulmonary primary bronchi, and the proximal portions of the intrapulmonary primary bronchi contain cartilaginous rings.

Perry (1988) also reports that the internal topography of the bronchi of the right lung was a mirror image of the left reflected along the mediosagittal plane, whereas we observed considerable asymmetry. This is largely a function of differences in the anatomy and branching angles of the cranial medial bronchi between the left and right lungs and the position of the left and right cardiac lobes. It does not appear that body mass can account for the differences between the studies because the body masses of the specimens were similar in both studies.

Branching patterns

Descriptions of the conducting airways of birds and mammals have relied on terminology that relates in part to the degree of branching that has taken place. However, to fully understand the branching pattern requires detailed knowledge of the development of the airways (Metzger et al., 2008), which is lacking for crocodilians, and so this terminology can be misleading.

Comparison with Alligator mississippiensis

The overall similarity between the primary, secondary, and tertiary bronchi of Crocodylus niloticus and A. mississippiensis is striking (Figs. 11 and 12), suggesting similar genetic control underpinning the branching patterns of the major bronchi in Crocodylia. The anatomy and position of the CVB (D1) and D2-4 are distinctly similar in all specimens of C. niloticus and that of A. mississippiensis (Sanders & Farmer, 2012) (Figs. 11 and 12). The proximal M branches (bronchi) are also similar in both taxa. This may be due to the importance of these bronchi in maintaining the integrity of the aerodynamic valve. Another distinct similarity between A. mississippiensis and C. niloticus is the hook at the terminal end of the primary bronchus and the caudally extending saccular structure (see Fig. 3).

Figure 11 3D segmented models of the bronchial tree of two live specimens of Alligator mississippiensis (in situ), and three specimens of Crocodylus niloticus generated from µCT and medical grade CT, all in dorsal view. (A) The primary, secondary, and tertiary bronchi of a 2.8 kg A. mississipiensis; (B) the primary, secondary, and tertiary bronchi of a 11 kg A. mississippiensis; (C) the primary, secondary, and tertiary bronchi of a 0.5 kg C. niloticus (NNC9); (D) the primary, secondary, and tertiary bronchi of a 0.8 kg C. niloticus (NNC6); (E) the primary, secondary, and tertiary bronchi of a 0.9 kg C. niloticus (NNC5). Images not to scale. Color scheme: white, trachea and primary bronchi; mint green, cervicoventrobronchi (CVB); lime, D2; neon green, D3; aqua, D4; light aqua, D5; light blue, D6, periwinkle, D7; blue, laterobronchi; purple, caudal group bronchi (CGB); red, M1; neon pink, M2; medium pink, M3; light pink, M4; pale pink, M5; pale purple-deep pink-purples, M6-8; yellow-gold, cardiac lobes.

Figure 12 3D segmented models of the bronchial tree of two live specimens of A. mississippiensis (in situ) and three cadaveric specimens of Crocodylus niloticus generated from µCT and medical grade CT, all in left lateral view. (A) The primary, secondary, and tertiary bronchi of a 2.8 kg A. mississippiensis; (B) the primary, secondary, and tertiary bronchi of a 11 kg A. mississippiensis; (C) the primary, secondary, and tertiary bronchi of a 0.5 kg C. niloticus (NNC9); (D) the primary, secondary, and tertiary bronchi of a 0.8 kg C. niloticus (NNC6); (E) the primary, secondary, and tertiary bronchi of a 0.9 kg C. niloticus (NNC5). Images not to scale. Color scheme: white, trachea and primary bronchi; mint green, cervicoventrobronchi (CVB); lime, D2; neon green, D3; aqua, D4; light aqua, D5; light blue, D6, periwinkle, D7; blue, laterobronchi; purple, caudal group bronchi (CGB); red, M1; neon pink, M2; medium pink, M3; light pink, M4; pale pink, M5; pale purple-deep pink-purples, M6-8; yellow-gold, cardiac lobes.

The major differences between the two taxa are subtle, yet suggestive of which pulmonary characters within Crocodylia may be plastic and which are conserved and thus putatively ancestral for the group. Crocodylus niloticus consistently has both more D and M branches than the alligator, as well as significantly more caudal group bronchi (CGB). The CGB are also evenly distributed around the non-cartilaginous intrapulmonary primary bronchus in C. niloticus, whereas they are primarily restricted to the ventrolateral surface in Alligator mississippiensis. Farmer & Sanders (2010) identified some large bronchi arising from the dorsal surface of the primary bronchus in the alligator as CGB. However, we consider that these are actually caudal dorsobronchi due to their large ostia, overall morphology, and dorsocranial orientation. Aside from the number of bronchi, the most visible difference between the two taxa is the topography of the tertiary bronchi. In C. niloticus the major tertiary branches of the CVB and D2-4 form an anatomical topology similar to that of the avian laterobronchi; (i.e., they run ventrally and branch into a multichambered sac-like structure). In A. mississippiensis, the major tertiary branches of the first four secondary bronchi are tube-like and run cranially in unison with their parent branches. Some alligators also have accessory branches emanating from the CVB that have not been observed in C. niloticus.

Our observations lead us to infer that certain aspects of the crocodilian bronchial tree are more plastic than others, particularly the number of secondary bronchi and the morphology of the tertiary bronchi. The origin and anatomy of the ventrobronchus (the CVB) and the dorsobronchi appear to be key features in the aerodynamic valves in both the Nile crocodile and the American alligator, whereas the tertiary bronchi and the CGB are more variable in form. Due to the functional relationship between the CVB and the dorsobronchi, we predict that this morphology will be present in other crocodilians, although Gavialoidea remains an important, although difficult to access, target of study.

Proposed homologies with the avian lung

A discussion of hypothesized homologies between the embryonic and juvenile lungs of the American alligator and the chicken were presented in great detail by Sanders & Farmer (2012) and so an extensive review will not be given here. However, a few anatomical characters that may be important to maintaining unidirectional flow will be discussed here as well as certain aspects of the lung relevant to the evolution of the archosaurian respiratory system. A diagrammatic comparison of the proposed homologies in the crocodilian lung and the avian lung are presented in Fig. 13.

Figure 13 Diagrammatic representations of the crocodilian (A) and avian (B) lungs in left lateral view with colors identifying proposed homologous characters within the bronchial tree and air sac system of both groups. The image of the bird is modified from Duncker (1971). Abbreviations: AAS, abdominal air sac; CAS, cervical air sac; CRTS, cranial thoracic air sac; CSS, caudal sac-like structure; CTS, caudal thoracic air sac; d, dorsobronchi; GL, gas-exchanging lung; HS, horizontal septum; IAS, interclavicular air sac; L, laterobronchi; NGL, non-gas-exchanging lung; ObS, oblique septum; P, parabronchi; Pb, primary bronchus; Tr, trachea; v, ventrobronchi.

The cervical ventral bronchus (CVB) - Sanders & Farmer (2012) proposed that the CVB of the alligator was homologous to the embryonic avian cervical air sac, the lateral moiety of the interclavicular sac, and the first ventrobronchus, based upon developmental data in crocodilians (Broman, 1939), the chicken (Locy & Larsell, 1916), and anatomical data in the juvenile alligator, as well as direct measurements of flow in the CVB that correspond to flow patterns in adult birds in these regions of the lung (Brackenbury, 1971; Brackenbury, 1972; Brackenbury, 1987). These data are supported by both the anatomy and the direction of airflow observed in the CVB of Crocodylus niloticus (Figs. 8, 10 and 13).

The dorsobronchi - The dorsobronchi in Alligator mississippiensis and in Crocodylus niloticus are anatomically similar to those of the bird and airflow patterns are similar in that they flow caudally to cranially in both taxa (Fig. 13; Duncker, 1971; Farmer & Sanders, 2010).

The parabronchi - The anastomosing tubular structures connecting the dorsobronchi with the CVB in Crocodylus niloticus were originally identified as possible homologues of the avian parabronchi by Huxley (1882). The structures found in C. niloticus are in both the same anatomical position as the parabronchi in birds and serving the same function (connecting the dorsobronchi with the ventrobronchi and facilitating flow between the former and the latter) (Figs. 7, 10 and 13). Parabronchial diameter in birds varies according to species, ranging from 0.05 mm in diameter in hummingbirds to 2 mm in coot, chicken, king penguin, and mute swan (Duncker, 1971). The structures identified in the 0.5 kg and 0.58 kg specimens of C. niloticus examined for this study were approximately 1.8 mm in diameter, within the size range found in birds.

The caudoventral saccular region - The primary bronchus continues caudally, widening into a canal that gives rise on its ventral and lateral surfaces to numerous small ostia that open into sac-like structures making up the ventral and lateral portion of the lung. Further caudally, the primary bronchus balloons into a variable number of sac-like chambers that make up the most caudal aspect of the lung (Fig. 3). Both the ventral and lateral structures and the caudal structures are less vascularized and less morphologically suited to gas exchange than the dorsal regions (Perry, 1990) occupied by the dorsobronchi (Fig. 13). As proposed for Alligator by Sanders & Farmer (2012), these bronchi are likely homologous to the avian laterobronchi and the caudal thoracic and abdominal air sacs (Fig. 13). This morphology is also present in Alligator mississippiensis (Sanders & Farmer, 2012).

In chickens, the abdominal air sacs are expansions of the distal tip of the primary bronchus, whereas the other air sacs all arise from secondary bronchi (Locy & Larsell, 1916). In some adult birds, such as chickens, this morphology becomes greatly exaggerated as the abdominal air sac develops into a massive sac. In contrast, in other birds such as the kiwi, the abdominal air sac is so diminutive that it is difficult to find. Indeed Richard Owen proposed it was lacking altogether, but Thomas Huxley identified a vestigial nub (Huxley, 1882).

While there is indeed evidence to suggest homology, the saccular regions of the Nile crocodile are anatomically very different from the air sacs of birds. In birds there are one or more narrow ostia that lead from either the secondary bronchi or the primary bronchus (in the case of the abdominal sac) into the sac. The caudal thoracic sacs and the cranial sacs are contained within the cavum subpulmonale, which is bounded dorsally by the horizontal septum (the avian diaphragm) and ventrally in part by the oblique septum (Huxley, 1882; Duncker, 1971), whereas the abdominal sac is bounded by the oblique septum and occupies the lateral regions caudal to the cavum subpulmonale (Fig. 13). These septa are formed by the invasion of the pulmonary fold by the lateral thoracic air sacs, which split apart the pulmonary fold into two layers so that the two separate septa are formed: the horizontal (sacco-pleural membrane = avian pulmonary diaphragm) and the oblique septum (thoracoabdominal diaphragm or sacco-peritoneal membrane). In the crocodile, the saccular regions are not projections from the rest of the lung so that the entire lung is readily dissected from the surrounding organs and has an outer contour that is smooth and loaf-like (Figs. 1B and 2A). Importantly, although lacking in crocodilians, pulmonary diverticula containing little or no gas-exchange parenchyma and that are reminiscent in ways of avian air sacs exist in numerous squamates, including varanids and chameleons (e.g., Milani, 1894; Perry, 1998). In summary, although we homologize the saccular caudoventral regions of the lungs of the Nile crocodile with the laterobronchi and caudal air sacs of birds and the first dorsal bronchus of the Nile crocodile with the cervical air sac and lateral moiety of the interclavicular sac as well as the first avian ventrobronchus, we emphasize that these saccular regions are very distinct in birds and crocodilians, just as the wing of a bird is distinct from the arm of a human. Both are forelimbs that arise embryologically from the same tissues in the same region of the embryo, but they form into distinct structures as they mature.

Patterns of airflow

As in extant birds (Butler, Banzett & Fredberg, 1988; Wang et al., 1988) and Alligator mississippiensis (Farmer & Sanders, 2010; Sanders & Farmer, 2012), no mechanical valves or sphincters were found in Crocodylus niloticus. The conversion of the tidal flow that is entering and leaving the trachea to unidirectional airflow within the lungs in Crocodylus niloticus appears to be produced by the geometry of the primary and secondary bronchi (Figs. 9 and 10). Our data suggest it is this bronchial arrangement that generates the inspiratory and expiratory aerodynamic valves. This has been verified in both C. niloticus and A. mississippiensis via gross dissection, µCT and medical grade CT. In the avian lung, unidirectional airflow in the secondary bronchi occurs via an inspiratory valve in which flow patterns are maintained by the branching angles of the primary and secondary bronchi and convective inertial forces (Butler, Banzett & Fredberg, 1988). A similar and perhaps homologous mechanism is likely functioning in C. niloticus. The hairpin turn of the CVB off of the primary bronchus, along with the anastomosing parabronchi linking the CVB to the dorsobronchi, provides an architectural arrangement similar to that in the bird (Fig. 10) and this geometry is consistent with the mechanism for the aerodynamic valves proposed by Hazelhoff (1951) and by Butler and colleagues (Butler, Banzett & Fredberg, 1988; Wang et al., 1988). Also, the close proximity of the cardiac bronchi has been hypothesized to be involved with unidirectional cardiogenic airflow in Alligator mississippiensis (Farmer, 2010) and due to anatomical similarities a similar mechanism may be occurring in C. niloticus. The specific topography of the saccular regions of the lungs probably play little, if any, role in the crocodilian aeorodynamic valve, which is similar to the situation in birds.

In birds, the air sacs serve as ventilatory bellows and storage reservoirs that move air through the primary and secondary bronchi; however, the shape of these sacs do not play any known role in generating the direction of flow within the gas-exchanging portion of the lung. Brackenbury, Darby & El-Sayed (1989) demonstrated by experimentally occluding the thoracic sacs (= the cranial and caudal thoracic sacs) in adult White Leghorn chickens that these sacs had almost no effect upon the ability of the bird to regulate intrapulmonary airflow during both resting conditions and exercise. In a second set of experiments Brackenbury & Amaku (1990) occluded both pairs of thoracic sacs and the abdominal air sacs ( ≈ 70% of tidal volume) resulting in a diminished respiratory function but no effect on inspiratory valving, thus demonstrating these air sacs collectively play little role in flow patterns within the gas exchanging lung. That is, aerodynamic valving is not dependent on the presence, location or topography of thoracic (cranial and caudal) and abdominal air sacs.

Origin and evolution of unidirectional airflow

The data of the bronchial topography and patterns of airflow in Nile crocodiles indicate that key features of the respiratory system are shared with both American alligators and birds. The most parsimonious interpretation of these observations is that these features were present in the common ancestor of birds and crocodilians (Archosauria) and were retained in both lineages. These observations are important for several other reasons. They corroborate the hypothesis that the topography of the bronchial passages themselves form the aerodynamic valves that transform the tidal flow that is entering and leaving the trachea into unidirectional flow within the lung (Dotterweich, 1936; Hazelhoff, 1951; King, 1966; Brackenbury, 1971; Duncker, 1971; Brackenbury, 1972; Brackenbury, 1979; Brackenbury, 1987; Butler, Banzett & Fredberg, 1988; Wang et al., 1988). Furthermore, they demonstrate that avian style air sacs are not required for flow-through lungs as has been proposed (O’Connor & Claessens, 2005) or for unidirectional flow (see below for a discussion of the difference in terminology of “flow-through” lung, as used by O’Connor & Claessens (2005) and in the physiological literature). This raises the interesting question of the selective driver(s) that might have originally favored the evolution of avian air sacs (Farmer, 2006). For example, avian air sacs may play important roles in sound production (Plummer & Goller, 2008), controlling pitch and roll during slow speed aerial maneuvers (Farmer, 2006), or preventing hypocapnia during panting (Brackenbury, 1971; Maina & Nathaniel, 2001).

Terminological turmoil

Comparative physiologists have long used the phrase “unidirectional flow in the lungs” and “flow-through lungs” synonymously. For example, Brown, Brain & Wang (1997) state:

“The parabronchi complete an airway loop from the caudal primary bronchus to the cranial primary bronchus (via secondary bronchi), through which a unidirectional stream of fresh gas flows (Figs. 1 and 4). That is, birds have a flow-through lung (parabronchi) in contrast to the tidal ventilation that occurs in mammalian alveoli.” pg. 189

However, the phrases were not used synonymously in the 2005 publication by O’Connor and Claessens (pers. comm, 2013), thus creating confusion because this deviation from standard usage was not stated in the text. The authors stated (O’Connor & Claessens, 2005: pg. 255) “Although our model does not predict the specific type of intrapulmonary air flow in non-avian theropods (unidirectional vs. bidirectional), it does establish both pulmonary and skeletal prerequisites required for flow-through ventilation.” Our understanding is that O’Connor and Claessens did not mean to communicate that they thought unidirectional flow was present inside the lungs of theropod dinosaurs, but rather they meant that these animals had air flowing through the lung from one region differentially to another region, thus creating a new definition for the term “flow-through lungs”. By this definition, snakes and other squamates, turtles, amphibians, and lung-breathing fish could be considered to have “flow-through” lungs as they all have regional differences in the distribution of gas exchanging parenchyma within the lung (e.g., Milani, 1894; Milani, 1897; Perry & Duncker, 1978; Perry, 1989; Perry, 1998; Wallach, 1998). However, it would be unusual for respiratory physiologists to refer to these lungs as “flow-through”, as this term is normally reserved for the avian lung and means that the air flows unidirectionally through the parabronchi. Here, we exclusively use the standard definition.

Pneumaticity, air sacs, lung efficiency, metabolism, and patterns of air flow in fossil taxa

There has been considerable attention given to reconstructing avian-like air sacs in extinct archosaurs based upon patterns of pneumaticity (e.g., Benson et al., 2011; Butler, Barrett & Gower, 2012; Claessens, O’Connor & Unwin, 2009; O’Connor & Claessens, 2005; O’Connor, 2006; Sereno et al., 2008; Wedel, 2003; Wedel, 2007; Wedel, 2009; Witmer, 1997). It has been suggested that regions of the postcranial axial skeleton are invariably and unambiguously pneumatized by specific air sacs or lungs in birds, and that these patterns of pneumatization can therefore serve as osteological correlates to interpret and reconstruct the presence of specific air sacs in extinct archosaurs O’Connor & Claessens, 2005; O’Connor, 2006; O’Connor, 2009; Wedel, 2006; Wedel, 2007; Wedel, 2009. Furthermore, these patterns of pneumaticity have been purported to be indicative of patterns of airflow throughout the bronchial tree of extinct archosaurs, the ‘efficiency’ of their lungs, their metabolic capacities, and their thermoregulatory strategies (e.g., Benson et al., 2011; O’Connor & Claessens, 2005; Wedel, 2003). The term ‘efficiency’ normally refers to the energy obtained out of a system per unit of energy put in, yet pneumaticity itself has never been quantitatively linked to measures of actual efficiency, whether considered to be oxygen extraction (Dejours, 1981) or ventilatory equivalent (rate of ventilation/rate of oxygen consumption). A full review of this body of literature is not possible here, but we briefly discuss several recent studies. Based on the presence or absence of postcranial pneumaticity and an array of vertebral laminae and fossae, Butler, Barrett & Gower (2012) proposed that pulmonary air sacs were present in the common ancestor of Ornithodira and were subsequently lost in ornithischian and other members of the clade. Benson et al. (2011) concluded that pneumatization in non-volant maniraptoran nonavian theropods evolved in association with an elevated metabolic rate and “high-performance” endothermy. Wedel (2007) and Wedel (2009) concluded that air sac-driven pulmonary ventilation was ancestral for Saurischia based on the presence of vertebral pneumaticity in various sauropods and theropods. Other osteological features, besides pneumaticity, have been used to try to reconstruct pulmonary anatomy. For example, Schachner, Lyson & Dodson (2009) and Schachner et al. (2011) used the costal and vertebral anatomy of a number of theropods and other dinosauriform archosaurs, as well as the rib anatomy of selected extant taxa, to retrodict the presence of dorsally immobile avian-like lungs. Many other studies have also tried to use pneumaticity to sort out respiratory anatomy and the presence or absence of specific patterns of flow in the lungs of extinct vertebrates (e.g., O’Connor & Claessens, 2005; O’Connor, 2006; O’Connor, 2009; Wedel, 2006; Wedel, 2007; Wedel, 2009). Yet postcranial pneumaticity has been purported to be equivocal evidence at best for patterns of air flow, lung efficiency, thermoregulatory strategies, and exercise capacities because pneumaticity has no known function in respiration or gas exchange (Farmer, 2006). Here we have shown that Nile crocodiles neither have postcranial pneumaticity nor air sacs and yet have lungs with truly flow-through ventilation. Hence unidirectional ventilatory flow (a flow-through lung in physiological terms) is possible in an ectothermic animal without pneumaticity and without air sacs. This emphasizes that bronchial anatomy, air sac anatomy, and ventilatory patterns can be decoupled from each other in archosaurs and should not be presumed to be correlated in simple ways.

We thank staff of the Structure & Motion Laboratory for assistance, as well as radiographic support and advice from Renate Weller and Victoria Watts, La Ferme aux Crocodiles; particularly Samuel Martin, Eric Fernandez, and Adrien Tomas; for their provision of crocodile cadavers for usage in this study, and Jason Bourke for assistance with Avizo. We thank P. O’Connor and M. Wedel for two very helpful reviews that greatly improved the manuscript, and we thank editor Andrew Farke for his guidance. We would also like to thank R. M. Elsey of the Rockefeller Wildlife Refuge in Louisiana for providing the American alligators for this study. E.R.S. is an American Association of Anatomists Scholar.

Additional Information and Declarations

Competing Interests

Author Contributions

Animal Ethics

John Hutchinson is an Associate Editor for PeerJ. The authors declare no other competing interests.

Emma R. Schachner conceived and designed the experiments, performed the experiments, analyzed the data, wrote the paper.

John R. Hutchinson performed the experiments, contributed reagents/materials/analysis tools, wrote the paper.

CG Farmer conceived and designed the experiments, performed the experiments, analyzed the data, contributed reagents/materials/analysis tools, wrote the paper.

The following information was supplied relating to ethical approvals (i.e. approving body and any reference numbers):

Approval for this study was granted from the University of Utah Institutional Animal Care and Use Committee (IACUC), protocol number 10-12003.

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
