# Peer review of "Pulmonary anatomy in the Nile crocodile and the evolution of unidirectional airflow in Archosauria"

_PeerJ, doi:10.7717/peerj.60_

## Round 0.1 · original submission · Major Revisions

Firstly, thank you for your patience during the protracted review period. The reviewers were abundantly enthusiastic about the novelty and importance of the data presented here, but had some reservations about the broader context in which the results are interpreted and presented. Thus, the reviewers have made a number of comments and suggestions for how to improve the manuscript, and I recommend that you consider all as you formulate revisions. Some specific recommendations include:

1) Although a continuous-loop/unidirectional flow model is suggested here for Crocodylus niloticus, both the reviewers and I were hard-pressed to see an explicit formulation of the model. As currently written, the description and illustration of the flow is rather oblique in nature - in other words, what is the precise path that a bolus of air takes as it travels through the respiratory system? In revision, this issue must be addressed. This may be through additional text and/or an additional schematic figure (strongly recommended).

2) Both reviewers suggest a refinement of the characterization of the function of avian abdominal air sacs, particularly noting that these air sacs are essential for avian respiration (even though the text implies otherwise, in some readings). Thus, the characterization of the role of air sacs--in birds in particular--should be carefully worded in the revised manuscript.

3) "Abdominal air sacs" in crocodilians are a notable feature that receive considerable attention in the discussion, but are scarcely included in the description or figures (and then, only under different terminology, as near as I can tell). Please augment the treatment of these air sacs (or their anatomical equivalents) in the main description and figures, perhaps by some additional explanation and labeling. It might also be helpful to reference the appropriate figures at the relevant point in the discussion, as an aid to the reader.

4) Both reviewers (and I) are in agreement that several major papers on the evolution of the archosaur respiratory system should be cited and worked into the manuscript (e.g., Butler et al., 2012, and other references mentioned by the reviewers). Please do so in your revision, as appropriate. Additionally, please note the important clarifications by O'Connor on what was specifically stated in his paper.

5) Reviewer O'Connor recommends close attention to how the terms "ventilation" and "inhalation" are used, to avoid any confusion.

6) Both the reviewers and I advise caution in how the results are applied to a broader understanding of pulmonary evolution in archosaurs. As noted by Reviewer Wedel, crocodilians are derived in many ways (behavior and anatomy), and caution should be used in extending conclusions outside of this group. The paper may benefit (optional for revision) from a more thorough consideration and mapping of the relevant homologies, to better get at what we do and don't know about the condition across archosaurs.
* * *
MINOR COMMENTS FROM THE EDITOR
The caption for figure 3 states, "For a detailed model of the anatomy see Figs. 3 and 4." Should it be "For a detailed model of the anatomy see Fig. 4."?

In a few places in the review PDF that I downloaded, the author names of papers cited mid-sentence are doubled up. For instance, "These observations are consistent with reports of Reese (Reese, 1915)". It could instead be listed as "These observations are consistent with reports of Reese (1915)."

On p. 14, it is stated that the abdominal air sac branches from the primary bronchus during Day 6 of embryonic development. This seems oddly specific, and I would suspect is taxon-dependent. Is this the case? Has the pattern only been described in domestic chickens? In any case, I would recommend a minor rephrasing of the sentence if necessary.

I am not an expert on archosaur lung flow, and would have benefited from a simple diagram showing the reconstructed flow of air through the lungs during a ventilation cycle. Although Figure 8 came close to this, it wasn't completely intuitive for me (even after several readings of the text).

·

Basic reporting

Basic Reporting
The material included in the manuscript adheres to most aspects of the PeerJ standards. A couple of comments: The introduction is too long and reticulate (see my specific comments on the MSWord document (with changes tracked) about overly wordy sections that are better suited for the discussion).

There are few statements made in the introduction that require better referencing (see MSWord document).

The figures (and supporting 3D characterization) in the paper are stunning, even with the sometimes sparse descriptive text to accompany them. This may be a situation when the picture is better than what could be done with text! Kudos to the authors for attempting to characterize the morphology of tertiary-level bronchi (and beyond).

Experimental design

Experimental Design
The anatomical observations based on 3D reconstructions provide a solid foundation around which to develop the paper.

Validity of the findings

The results presented in this paper are robust (although see my specific comments on the associated MSWord file).

The discussion (at times) goes on into areas for which the collected dataset does not offer any specific constraint. I have indicated this in my comments on the MSWord file.

Additional comments

General Comments for Authors
See the associated MSWord document for a number of suggestions/edits, questions.

The following two sections need to be fleshed out in much more detail in the paper: I think it is interesting what you are proposing, but it is not clear from the specific anatomical description how these work to effect the continuous loop.

1. Small diameter anastomosing bronchi (Parabronchi)

2. Airflow patterns in the major secondary bronchi

In additon to the general comments regarding references in the MSWord documnet, this paper MUST clearly integrate the information from: Butler RJ, Barrett PM, Gower DJ (2012) Reassessment of the Evidence for Postcranial Skeletal Pneumaticity in Triassic Archosaurs, and the Early Evolution of the Avian Respiratory System. PLoS ONE 7(3): e34094. doi:10.1371/journal.pone.0034094


As the two papers are so clearly complementary--to not include the Butler et al (2012) paper into this work would be a major oversight.

·

Basic reporting

Page 3, lines 49-51. "Bird-like lungs have been proposed to have evolved in pterosaurs and non-avian theropods (O'Connor and Claessens 2005), and to have appeared first in non-avian coelurosaurs (Perry 1992). However, the recent discovery of unidirectional airflow in the lungs of alligators (Farmer 2010; Farmer and Sanders 2010) suggests these characters are far older."

This passage makes it sound as if Perry (1992) was the state of the art until 2010, and as if no-one had suggested unidirectional airflow for any extinct archosaurs outside of coelurosaurs before Farmer and Sanders (2010)--neither of which is accurate. In doing so, it skips over much relevant work on pterosaurs (e.g., Bonde and Christiansen 2003, Claessens et al. 2009), sauropodomorphs (e.g., Wedel 2009, Yates et al. 2012), and non-avian archosaurs in general (e.g., O'Connor 2009, Butler et al. 2012). Expanding this section to discuss how we got to our present understanding of archosaurian respiration and citing the relevant sources (the suggested references are not exhaustive) would improve the paper, first by putting the current work into its broader context, and second by giving credit to the numerous other workers who have explored the possibility of bird-like respiration in non-avian archosaurs.

Page 15, lines 354-355. "These data also suggest that the extensive air sacs found in extant birds, and those reconstructed for extinct non-avian theropods (O'Connor 2006; Wedel 2006; Wedel 2007; Wedel 2009)"

As discussed above, air sacs have also been reconstructed in pterosaurs, sauropodomorphs, and the ancestral ornithodiran, so it would be better to say, "and those reconstructed for extinct archosaurs". Please also cite Claessens et al. 2009, Butler et al. 2012, and Yates et al. 2012 here.

Experimental design

Page 5, Materials and Methods. Please report both masses and lengths for as many specimens as possible. Not all of us have enough experience with alligators and crocs to have a sense of how big 10kg, 2kg, and 0.5kg animals are, either absolutely or compared to one another. Also, the statement about IACUC approval is currently buried in a passage on a CT techniques and should be moved to the beginning of the whole section.

Results. The descriptions of the various bronchi are very clear and the illustrations are fantastic--not just clear but beautiful. However, more quantitative data would be very useful. (1) How many parabronchi are present? It's not clear from either the text or the figures. Even a ballpark number would be helpful. (2) What is the internal diameter of all of these passages (or minimum diameter in cases where it varies along the tube)? (3) For the passages that are involved in generating the unidirectional airflow, how does the diameter of the tubes compare to that of homologous structures in birds of equivalent mass (especially the parabronchi)? (4) Given that the sample includes animals ranging from 0.58-14.6kg, are there any important changes related to size/age? Are the lungs of larger crocodilians just scaled-up copies of the lungs of babies? Does the number or diameter of the parabronchi change?

Validity of the findings

Page 3, lines 56-59. "In this scenario, the unidirectional airflow found in birds, which appears to facilitate their ability to fly in hypoxic conditions (Meyer et al. 1981), is an exaptation, having originally served their distant ancestors in a completely different role, that of facilitating gas exchange during apnea."

I understand the evidence for cardiogenic airflow in Alligator and find it convincing. The demonstration of unidirectional airflow in extant crocodilians is also rock-solid, as is the inference that unidirectional airflow is primitive for archosaurs. However, that does not mean that the ancestral archosaur used unidirectional airflow to keep air moving around in the lungs during apnea, as suggested here. Cardiogenic airflow is advantageous for extant crocs because they are aquatic ambush predators, but the ancestral archosaur was not, nor were many of the linear ancestors of crocodilians among the croc-line archosaurs. Indeed, if sphenosuchians are paraphyletic as suggested by Nesbitt (2011) and other recent analyses, then the direct ancestor of all crocodyliforms was a small, long-legged, terrestrial animal, probably cursorial and possibly insectivorous. We can confidently infer that that ancestral croc had unidirectional airflow in its lungs, but holding its breath underwater was not an important part of its behavioral repertoire. The larger point that I am building up to is that it is a mistake to assume that extant crocs are representative of the primitive archosaurian condition. Crocs and birds are both derived, in very different directions, from the ancestral archosaur. In fact, cardiogenic airflow during apnea is even more likely to be an exaptation than is remaining active (flying, or running for non-volant animals) in hypoxic conditions, given that the radiation of archosaurs occurred when global atmospheric oxygen levels may have been lower than they are today.

I strongly recommend that you take all of this into account in revising this passage and throughout the remainder of the manuscript. It is particularly important that you cite Nesbitt (2011) and emphasize that extant crocodilians are quite different from ancestral crocodyliforms and ancestral archosaurs in general in both gross morphology and ecology, so that readers don't fall into the trap of assuming that extant crocs are representative of whole crurotarsal radiation.

Page 13, Evolution of the abdominal air sac. This whole section is extremely interesting, and mind-blowing in its implications. Beyond the bronchi and the abdominal air sac, are there any other regions of the crocodilian lung that seem to be homologues of specific parts of avian lungs and air sacs? I'm asking out of sheer curiosity. Whether the answer is yes or no, it would be worth making it explicit in the text.

Page 14, Patterns of airflow. One thing I still don't understand is the path of the air taken in during a single inspiration. In birds inspired air usually goes into the posterior air sacs on the first inhalation, then into the lungs, then into the anterior air sacs on the second inhalation, then out on the second exhalation. Is the pattern the same for crocs, or do they get it all over with in a single inspiratory/expiratory cycle? If it's the same, what parts of the lung are functioning like the anterior air sacs of birds? If it's different, how is it different, and can you make any inferences about how (mechanically) or why the avian pattern was established? It might be helpful to add a very simple diagram showing where the air is going on each inspiration and expiration, of the sort that Schmidt-Nielsen used to illustrate his classic papers on avian breathing.

Page 15, lines 349-357. "it was the discovery of unidirectional airflow in the alligator that unambiguously demonstrated that extrapulmonary air sacs are unnecessary for the maintenance of unidirectional airflow in the archosaurian lung...these data suggest that the extensive air sacs of birds...performed an unknown and as of yet untested biological role other than respiration."

I agree that _extrapulmonary_ air sacs are not necessary for unidirectional airflow. I disagree that this means that avian air sacs evolved to perform some other function. As you argue just a few paragraphs earlier, birds and crocs both have abdominal air sacs, it's just that those of birds are extrapulmonary and those of crocs are only extrabronchial. Presumably the abdominal air sacs serve as reservoirs of air for the rest of the lung in both cases. And you cannot disconnect the lungs of a bird from the air sacs and still have a functional organism; in birds, the air sacs most definitely are necessary for the maintenance of unidirectional flow, because the lungs have no ventilatory ability of their own. Both birds and crocs have unidirectional airflow but birds have an even more heterogeneously partitioned respiratory system (to use Perry's terminology) than do crocs, and they are also endothermic and primitively (for the crown clade) volant. It seems intuitive that the further segregation of the respiratory system into ventilators (air sacs) and gas exchangers (the parabronchial portion of the lung) in birds is related to their high BMR and high activity levels. That hypothesis seems at least as reasonable as the suggestion that the extrapulmonary air sacs of birds and other archosaurs evolved for sound production or locomotion, and it has a long history in the literature (e.g., various papers by Perry), so I am curious why it is not mentioned.

The idea that extrapulmonary air sacs initially evolved to lighten the skeleton can be safely discarded; the evolution of air sacs almost certainly preceded the evolution of postcranial skeletal pneumaticity (Wedel 2007, 2009, Yates et al. 2012), and in the earliest saurischians with postcranial pneumaticity the amount of mass removed from the skeleton was negligible (Wedel 2007).

Additional comments

Papers cited in the other sections of the review:

Bonde N, Christiansen P (2003) The detailed anatomy of Rhamphorhynchus: axial pneumaticity and its implications. In: Buffetaut E, Mazin J-M, editors. Evolution and paleobiology of pterosaurs. London: Geological Society. pp. 217–232.

Butler, R.J., Barrett, P.M., and Gower, D.J. 2012. Reassessment of the evidence for postcranial skeletal pneumaticity in Triassic archosaurs, and the early evolution of the avian respiratory system. PLoS ONE 7(3): e34094. doi:10.1371/journal.pone.0034094

Claessens LPAM, O'Connor PM, Unwin DM (2009) Respiratory Evolution Facilitated the Origin of Pterosaur Flight and Aerial Gigantism. PLoS ONE 4(2): e4497. doi:10.1371/journal.pone.0004497

Nesbitt, S.J. (2011). "The early evolution of archosaurs: relationships and the origin of major clades". Bulletin of the American Museum of Natural History 352: 1–292. doi:10.1206/352.1

O’Connor, P. M. Evolution of archosaurian body plans: skeletal adaptations of an air–sac–based breathing apparatus in birds and other archosaurs. J Exp Zool 311A:629-646 (2009).

Yates, A.M., Wedel, M.J., and Bonnan, M.F. 2012. The early evolution of postcranial skeletal pneumaticity in sauropodomorph dinosaurs. Acta Palaeontologica Polonica 57(1):85-100. doi: http://dx.doi.org/10.4202/app.2010.0075

---

## Round 0.2 · Minor Revisions

Thank you for your detailed consideration of comments on the previous version of the manuscript. As a result, the paper is greatly improved--the additional figures are a particularly useful addition, and I suspect will be utilized frequently by many in the field. In light of the extensive revisions required for the previous draft, I elected to send it for one final round of reviews (to the previous two reviewers). They have made some comparatively minor suggestions (relative to the previous round), and I am confident that you will be able to address them with ease.

At this point, both reviewers continue to request a greater engagement with previous literature. Although the center of the paper is a fantastic and novel anatomical description of crocodilian lungs, the discussion over archosaur pulmonary evolution should more fully incorporate previous work (e.g., that of Butler et al., 2012). . .this is the primary theme of the new reviews. Indeed, some of the revisions have removed previously helpful discussions of this past week. I am beginning to recognize that this may represent some real philosophical differences between the authors and the reviewers, but I also think at least a _slightly_ more expanded incorporation of some of the previous literature is warranted, regardless of any philosophical (or scientific) disagreements.

Although their full comments are provided at the end of this email, I summarize the reviewers' statements with some commentary and clarification as follows:

1) Reviewer 2 (Wedel) is somewhat concerned over the near-complete elimination of the portion of text (original p. 15, lines 354-355) on extensive air sacs in extinct archosaurs (originally non-avian theropods, but suggested by Reviewer 2 to be expanded to include other taxa for which these air sacs are reconstructed with confidence), and how they relate to respiratory evolution. Reviewer 1 (O'Connor) echoes similar concerns. Although I understand your rebuttal that the primary focus of the manuscript is not on reconstructed air sacs, the average reader who is not familiar with the precise nuances of the issue may be confused by the omission (indeed, I was!)--and there are enough mentions of air sacs still in the discussion to warrant the inclusion of this literature in a meaningful way. As a compromise, please give strong consideration to _briefly_ addressing these papers dealing with air sacs, and noting (if you truly consider it to be the case) that air sacs are a separate issue from the bronchial anatomy, as you have done in your rebuttal. After all, air sacs in chameleons, etc., are already mentioned. This could be done as a few lines (perhaps at the point in version 1 of the manuscript), and I think would be enhance your contention that the emphasis of the paper is on bronchial structure/flow rather than air sacs, as well as satisfy those who want to see more on air sacs. Furthermore, it would advance a more holistic view of the whole archosaurian respiratory system (an important aspect of this paper), rather than a purely air sac-centric or lung-centric viewpoint. Finally, other parts of the revised discussion talk about evolutionary origins of air sacs (e.g., p. 18, line 425: "development of saccular regions/air sacs is likely not an avian autapomorphy." - postcranial pneumaticity in non-avian theropods, theropods, and pterosaurs establishes this beyond a doubt!). This would be a perfect juncture to briefly note the "reconstructed air sacs in fossil archosaurs" issue (and indeed, the omission of such reference to the fossil record at that point seems a little odd). Again, this does not have to be pages and pages of text--just a brief mention.

2) Based on the rebuttal, as well as the email exchange from Reviewer 1 (O'Connor), there still seems to be differing interpretations of flow-through vs. unidirectional ventilation between you and this reviewer. I would suggest that the most appropriate way to deal with this is to acknowledge the differences of interpretation and definition (and cite appropriate literature from both "sides"). A few sentences would suffice.

Thank you for your attention to these details. . .as noted, these are comparatively minor (and easily addressed) sidelines to the discussion on what is a very solid piece of anatomical work.

·

Basic reporting

The basic reporting in the contribution (R1) is robust and the revised version of the paper takes into account most of the suggested comments by the reviewers and editor.

Experimental design

Appropriate for the study at hand.

Validity of the findings

The anatomical and physiological findings in the contribution remain solid.

Stats = NA

Additional comments

Emma, Colleen, and John:

Nice work addressing the comments on the original submission. I think you have handled most of the nuts-and-bolts issues appropriately. Also, the additional figures are very nice to see.

The additional components/discussion on abdominal air sac/ventrocaudal saccular regions is appropriate. However, there still seems to be a disconnect between this work and previous research on the topical area (e.g., inferences related to air sacs [ventilators] in archosaurs; Butler et al., 2012).

You are specifically generating homology assessments/statements about the abdominal air sacs in birds and the ventrocaudal saccular region of the lung in crocodyliforms...[lines 425-430]...and it also clear that you state that “bronchial topography and patterns of airflow” [lines 453-454] are also primitive Archosaurian features. It would seem logical that both are essential/necessary to generate the flow patterns that you have documented.

If so, the following discussion item doesn’t seem to be the most parsimonious conclusion: “Furthermore, they [= observations] demonstrate that avian style air sacs are not required for flow through ventilation as has been proposed (O'Connor and Claessens 2005) or for unidirectional flow” [Lines 462-464]. Take away the air sacs in birds and what do you have in terms of ventilatory potential?

In a taxon (e.g., bird) with a rigid, non-compressible lung, it would seem that some type of saccular (i.e., ventilator) region would indeed be necessary to generate the airflow patterns (any airflow patterns). And given that some authors (e.g., Butler et al., 2012) have laid out the framework on the inferential basis for another, extraordinarily important part of the pulmonary system (i.e., air sacs) in archosaurs, and one that IS ESSENTIAL for ventilation in birds and perhaps crocs via the ventrocaudal saccular regions, it seems odd that you do not better integrate your model with it. Your work would be so much more significant if it would interface better with previous work.


Also, I am here including the email exchange that we had in January 2013 following the initial review. As this went into perhaps more detail than the submitted review on the topic of differentiating flow-through from uni-directional flow (sensu O’Connor and Claessens [2005], O’Connor [2006], I feel that making this part of the official review record at PeerJ is appropriate. I put this here more for tracking the general history of the review process, rather than for anything specifically related to my review of the revision.
* * *
Wed 1/16/2013 9:12 AM
Emma Schachner <eschachner@gmail.com>
Colleen Farmer <cg.frmr@gmail.com>; O'Connor, Patrick <oconnorp@ohio.edu>

Great. Glad you got the reviews.

The flow-through aspect that I am referring relates to an “hypothesized” initial differentiation of parenchymal density in the avian (or avian line archosaurs, or maybe even archosaurs more broadly based on the work you are doing) pulmonary system, whereby increasing parenchymal density near the dorsocranial end of the lung would serve as a focal region for gas exchange and the less (parenchymal) dense posterior regions of the lung would begin to serve as primarily ventilatory regions…the increasing parenchymal density in the exchanger would necessarily have a reduced compliance by comparison with the “ventilators”—at this point, I haven’t specified anything about bi-directional vs. uni-directional within the actual exchanger region (as I indicated in the first email, that was far beyond the scope of the dataset)—but assuming a plesiomorphic condition of bidirectional, tidal flow into the pulmonary apparatus (which is not heretical when you consider an aspiration [i.e., negative pressure inflow] systems more generally), it seems reasonable that the initial differentiation of parenchymal density (as indicated above) among the archosaurs may have existed in the following manner with regard to the exchanger: bidirection air-flow through the "exchanger" region during (i.e., into and through the gas exchange region into the posterior/caudal air sacs during inspiration AND back through the gas exchanger during expiration)--that is...flow-through system with bidirectional flow through the exchange region. But without any specific information on hand at the time to consider internal tubing (i.e., bronchial organization) in the basal archosaur condition (until now), and with one extant end-member (birds) with a non-compliant exchanger that is ventilated solely by compliant air sacs, it was seeking to provide a basic model on an hypothesized 'lineage" that provide the basic anatomical components that were at some point consolidated/tweaked into the condition we see in extant birds.
A fairly simple model indeed...and one that does not provide the specific anatomical details about what is happening in the "exchange region" of the lung, but one that appreciates the presence of a pulmonary apparatus that minimally has compliant components (air sacs) as part of the system. Now, a decade later, here we (i.e., you) are looking at intrapulmonary (anatomical and physiological) similarities among archosaurs more generally and we likely have a better chance of refining this type of modeling.
One point of refinement that is sometimes overlooked: air sacs are indeed not necessary for unidirectional airflow (as demonstrated by your and Colleen's work). But in the context of a rigid exchange region (like that in birds), air sacs are indeed required. In other words, it is not possible to have a more-or-less non-compliant exchanger without a mechanism to get air into and out of it. Right? I think if we all get on the same page about what uni-directional air flow (which is anatomically restricted to selective parts of the lung in both birds and crocs) vs. flow-through ventiliation (sensu O'Connor and Claessen, 2005, which is a whole pulmonary system model for airflow relative to an exchange region) are, then we can really start developing more robust models about inferred changes in pulmonary anatomy among archosaurs.
I would be happy to chat (phone or skype) with you (and Colleen) if you want to chew on this any more. Good luck with the revisions and I look forward to seeing more of this.

Pat

Patrick O'Connor, Ph.D.
Associate Professor of Anatomy
228 Irvine Hall
Department of Biomedical Sciences
Ohio University College of Osteopathic Medicine
Athens, OH 45701
ph/740-593-2110
fx/740-597-2778
Web: http://www.oucom.ohiou.edu/dbms-oconnor/
Rukwa Rift Basin Project: http://www.oucom.ohiou.edu/rukwa/index.htm



From: Emma Schachner [mailto:eschachner@gmail.com]
Sent: Tuesday, January 15, 2013 12:58 PM
To: O'Connor, Patrick
Cc: Colleen Farmer; hutch@rvc.ac.uk; afarke@webb.org
Subject: Re: review of Nile croc pulmonary anatomy paper

Hi Pat,
Yes I'm here. Thanks for the great review! I've been going through point by point and working on incorporating all of your suggested changes. I didn't want to send you questions until I had gone through it in detail.
One thing I am still a bit confused by however, is this 'flow through' lung concept. So are you just saying that air differentially flows from one region of the lung to a different region of the lung? Doesn't that occur in all lungs - sauropsids in particular? Or are you considering avian air sacs to be a separate organ from the lung? Developmentally they are secondary and tertiary bronchi
Cheers,
Emma

On Tue, Jan 15, 2013 at 8:12 AM, O'Connor, Patrick <oconnorp@ohio.edu> wrote:
Colleen:
Just making sure you received this and that Emma is also part of the email string (I haven’t heard anything from her)…she is the lead person on this, right?
Enjoy the lots and lots of snow that I have been hearing about in Utah.

Pat


Patrick O'Connor, Ph.D.
Associate Professor of Anatomy
228 Irvine Hall
Department of Biomedical Sciences
Ohio University College of Osteopathic Medicine
Athens, OH 45701
ph/740-593-2110
fx/740-597-2778

Web: http://www.oucom.ohiou.edu/dbms-oconnor/
Rukwa Rift Basin Project: http://www.oucom.ohiou.edu/rukwa/index.htm



From: O'Connor, Patrick
Sent: Thursday, January 10, 2013 7:05 AM
To: Colleen Farmer
Cc: eschachner@gmail.com; hutch@rvc.ac.uk; afarke@webb.org; O'Connor, Patrick

Subject: Re: review of Nile croc pulmonary anatomy paper

Hi Colleen and Emma:

Great to hear from you. And you are definitely in the right place to ski with your kids—the last time I went skiing was at Canyons following the SLC SICB. As an aside, I can only imagine what having two kids (only two years apart!) must have been like when they were little.

I hope that my comments are helpful…and feel free to shoot stuff by me in advance or as you are revising.
Regarding my concept of 'flow-through' ventilation (and why it is different with unidirectional air flow as in birds)…in the model the Leon and I developed, we were very careful to not specify 'intrapulmonary' flow patterns (as our data clearly did not speak to what was happening in the lung)--but to provide a basis for putting compliant components of the respiratory system (air sacs) on both the front end and back end of the exchanger. Of course we did this based on region-specific pneumaticity data and the established relationship of specific air sacs/diverticula to different parts of the skeleton in birds. This presence of skeletal pneumaticity also allows for the inference of a heterogenerously-partitioned pulmonary system (which is not a surprising given the range of heterogeneous pulmonary systems known among extant sauropsids anyways). By combining the soft-tissue inferences for air sacs AND what we gauged as potential kinematics throughout the thoracic skeleton (e.g., a relatively immobile region in the cranial end of the thorax vs. a relatively mobile region caudally based on "inferred relative" costovertebral movement), we modeled the presence of distinct exchanger (lung) and the cranial and caudal air sacs, whereby air during inspiration could be brought into and through the lung into the caudal sacs and pushed back through the exchanger during expiration (once again—this isn't specifying unidirectional air flow at all—it could still be tidally driven, bidirectional flow of air through the exchanger—FLOW THROUGH). Quoting from the O'Connor and Claessens (2005) paper…Although our model does not predict the specific type of intrapulmonary air flow in non-avian theropods (unidirectional vs. bidirectional), it does establish both pulmonary and skeletal prerequisites required for flow-through ventilation, A PLAUSIBLE EARLY STAGE in the evolution of the highly derived avian pulmonary apparatus. Of course this was 8/9 years ago when the project was completed and the paper submitted.
Move ahead to the work you guys (i.e., you, Kent, Emma) are and have been doing now…that is, providing a much better characterization of detailed pulmonary anatomy and function…this is allowing the field to evolve with respect to consideration the underlying similarities in gross pulmonary architecture (and function) among archosaurs more generally and to better formulate models about the evolution of the exchanger…something that I was never able to do in such detail. I really think that there is great potential for linking datasets/perspectives down the road (and not as part of your current paper)--this is where I was going with putting the symposium together for the Paris ICVM--and we did get more people to the table at that point. I think it is time to do so again.
By the way, the idea that bidirectional flow-through must have been present prior to the development of unidirectional flow-through (as in birds) is just that…an idea (which is why I am extremely explicit these types of inferences). My 2006 paper was able to further articulate this concept..see the discussion section--but I took it as far as I was comfortable doing so based on my primary focus at that time. I would love to spend more time thinking about this aspect of the model development with the lung-focused people and see what we might put together.
In any event, good luck revising the paper and let me know how I can help. I do want to get out to Utah for a visit at some point.
All best and hi to Dave.
Pat

Patrick O'Connor, Ph.D.
Associate Professor of Anatomy
228 Irvine Hall
Department of Biomedical Sciences
Ohio University College of Osteopathic Medicine
Athens, OH 45701
ph/740-593-2110
fx/740-597-2778
Web: http://www.oucom.ohiou.edu/dbms-oconnor/
Rukwa Rift Basin Project: http://www.oucom.ohiou.edu/rukwa/index.htm

From: Colleen Farmer <cg.frmr@gmail.com>
Date: Wednesday, January 9, 2013 8:08 AM
To: Patrick OConnor <oconnorp@ohio.edu>
Cc: "eschachner@gmail.com" <eschachner@gmail.com>, "hutch@rvc.ac.uk" <hutch@rvc.ac.uk>, Andrew Farke <afarke@webb.org>
Subject: Re: review of Nile croc pulmonary anatomy paper

Hi Pat,

I am glad to hear you have a two-year old to enjoy. I miss that stage. My kids are 13 and 11 and now we ski instead of building block towers.
Thank you for your thoughtful comments on the ms. I am sorry if I have not cited you correctly in the past. I misunderstood when you wrote flow-through and thought you meant unidirectional flow. I would appreciate it if you could explain to me what you mean when you use the term flow through ventilation.

Cheers,
Colleen
On Wed, Jan 9, 2013 at 12:33 AM, O'Connor, Patrick <oconnorp@ohio.edu> wrote:
Hi Emma, Colleen, and John:
cc: Andrew Farke, Editor PeerJ
Attached you will find my comments on your paper submitted to PeerJ. Sorry if this has taken a while to get back to you, but the review came at an extremely busy/crazy time. The combination of having a two-year old and the holidays work wonders for delaying things like reviews. Do not take that to mean your paper isn't important. On the contrary...this is exactly the kind of detailed anatomical work that is necessary for pushing these topics forward.
That anatomy in this great (stunning is the term I think I used in the review) and the implications of it equally exciting--I have provided many comments that I think will make the manuscript and interpretations that much more robust. More specifically, attached to this email are my comments on the MSWord file so that you can see the specific editorial bits and the questions/comments.
I would be happy to chat (phone or skype) if the event that you have any questions about my comments.
Happy New Year.
Pat
ps-please stop citing me as indicating that extrapulmonary air sacs are necessary for "unidirectional air flow"--this is not what is "explicitly indicated" in the O'Connor and Claessens (2005) paper. :) There is a difference between flow-through ventilation and unidirectional air flow. The latter is no doubt predicated on the former...but they are not the same thing.

End of email exchange
* * *
·

Basic reporting

MAJOR POINTS

Original manuscript: Page 15, lines 354-355. "These data also suggest that the extensive air sacs found in extant birds, and those reconstructed for extinct non-avian theropods (O'Connor 2006; Wedel 2006; Wedel 2007; Wedel 2009)"

My comment from the first review: As discussed above, air sacs have also been reconstructed in pterosaurs, sauropodomorphs, and the ancestral ornithodiran, so it would be better to say, "and those reconstructed for extinct archosaurs". Please also cite Claessens et al. 2009, Butler et al. 2012, and Yates et al. 2012 here.

Your response: We have not discussed the reconstructed air sacs. We primarily discuss our flow measurements, which are all in the trachea and the ventro and dorsobronchi, and we homologize regions of the crocodilian and avian respiratory systems.

I realize that you are coming at archosaur lung evolution from a different angle than the workers that I recommended you cite here. But you _are_ addressing archosaur lung evolution, and those papers are relevant. The solution is to actually engage with them, not cut those lines from the original manuscript.

To take the case of the Butler et al. (2012) paper in particular, in your rebuttal to Reviewer 1 you wrote,

“We have included the citation of Butler et al. to ensure there is no oversight. However, our paper does not deal with pneumaticity in either extant or extinct groups and so we do not think the Butler et al paper on pneumaticity in extinct lineages is integral to our measurements on the anatomy or patterns of air flow in the Nile crocodile.”

The point is not that Butler et al. is a paper on pneumaticity and yours is a paper on air flow in crocodile lungs. Both papers address homologies of the crocodilian and avian respiratory systems and evolutionary scenarios for the origin of the avian respiratory system, so Butler et al. (2012) is directly relevant to your own work. The fact that they are using a different line of evidence to arrive at similar conclusions makes their work complementary to yours. It may not be integral to your methods but it is certainly integral to your discussion and conclusions. So I agree with Reviewer 1 that you must _integrate_ Butler et al. (2012) into your discussion—not just give it a single drive-by citation in the introduction.

Page 17, lines 422-425: “Importantly, although lacking in crocodilians, pulmonary diverticula that are reminiscent of avian air sacs exist in numerous squamates, including varanids and chameleons (e.g., Milani 1894; Perry 1998). Hence the development of saccular regions/air sacs is likely not an avian autapomorphy.”

We already knew that air sacs were not an avian autapomorphy, because patterns of pneumaticity diagnostic for homologous air sacs have been found in pterosaurs (Claessens et al. 2009), sauropodomorphs (Wedel 2009, Yates et al. 2012), and non-avian theropods (O’Connor and Claessens 2005, Benson et al. 2011), and at least ambiguous evidence of pneumaticity has been identified in a host of other archosaurs, including some croc-line taxa (Butler et al. 2012). This is another point where not engaging with the relevant literature is actually seriously hurting your discussion, because it leads you to tentatively propose (“is likely not”) a hypothesis that is already robustly supported by published data, and widely accepted by other researchers interested in the history of archosaur respiration. Your work on crocodilians is new and exciting and valid, but it is weakened, not strengthened, by your refusal to integrate complementary work by others.

Page 3, lines 67-69: “The archosaur lung may be a key innovation that gave archosaurs a competitive advantage over the synapsids in niches that required highly aerobic metabolisms during the atmospheric hypoxia of the Triassic (Farmer 2010; Farmer and Sanders 2010).” Peter Ward has also discussed this idea in his books and papers, and it would be appropriate to cite him here.

MINOR POINTS

Page 2, line 26: “pterosaurs, and non-avian dinosaurs. The latter clade came to dominate…” Sorry to nitpick—and sorry I didn’t catch this the first time around—but ‘non-avian dinosaurs’ is a grade, not a clade. So maybe amend the opening of the second sentence to “The latter group” or “The latter clades”.

Page 3, lines 57-60: “the character of unidirectional airflow through open-ended, tubular gas-exchanging structures is far older than the ornithodiran lineage, and predates the evolution of avian style air sacs, having evolved in the common ancestor of the pseudosuchian and ornithodiran lineages.”

I would remove “is far older than the ornithodiran lineage”. Given that we have almost no definite basal avemetatarsalians, Archosauria is only one node deeper than Ornithodira, and given the apparently rapid pace of archosaur evolution in the Early and Middle Triassic, the common ancestor of Archosauria might not be that much older than the common ancestor of Ornithodira. So the “far older” clause implies a depth of history that is currently unsupported, and the sentence is still accurate and reads just fine without it: “the character of unidirectional airflow through open-ended, tubular gas-exchanging structures […] predates the evolution of avian style air sacs”.

Page 4, line 85. The long quote from Huxley that starts here and runs to page 5, line 103, would read better if it was set off by indenting, as a blockquote.

Pages 5-6, lines 121-124. “The clade Crocodylia is composed of at least two major lineages: the Alligatorioidea, which includes two species of alligators and the caimans, and the Crocodyloidea, which includes the Nile crocodile, Morelet’s crocodile, and dwarf crocodiles, as well as most other extant species.” This is oddly phrased—why mention the number of species for alligators but not the others? Also, could you not just say that Crocodyloidea includes all extant species of crocodiles? Finally, Alligatoroidea is misspelled. I think it would read better as, “The clade Crocodylia is composed of at least two major lineages: Alligatoroidea, which includes the two extant alligator species and seven extant caiman species, and Crocodyloidea, which includes the 13 extant species of crocodiles.”

Page 12, lines 288 and 303, and page 17, lines 393 and 426. Here you use the terms “ventrocranial” and “ventrocaudal”, whereas in the remainder of the manuscript you use the more common cranio-(dorsal, lateral, or ventral)—11 instances—and caudo-(dorsal, lateral, or ventral)—4 instances. I recommend using the more familiar ‘cranioventral’ and ‘caudoventral’.

Experimental design

No comments

Validity of the findings

Page 19, lines 461-468:

“Furthermore, they demonstrate that avian style air sacs are not required for flow through ventilation as has been proposed (O'Connor and Claessens 2005) or for unidirectional flow. This raises the interesting question of the selective driver that might have originally favored the evolution of avian style air sacs (Farmer 2006). For example, avian style sacs may play important roles in sound production (Plummer and Goller 2008) or controlling pitch and roll during slow speed aerial maneuvers (Farmer 2006). Hence the mystery of the potential selective drivers for the evolution of these intriguing and poorly understood structures remains unresolved.”

This section is still problematic. Although much of the wording has been modified (e.g., substituting “avian style air sacs” for “extrapulmonary air sacs”), the central problems remain.
1. Air sacs, whether extrapulmonary as in birds or merely extrabronchial as in crocodilians, do contribute to unidirectional airflow by ventilating the gas exchanging portion of the lung during exhalation (as you show for both birds and crocodilians in Figure 10). Perhaps a bird lung with no air sacs, or a croc lung with the non-gas-exchanging portions excised, could still generate unidirectional flow on inspiration. But it could not do so on expiration, because there would be no air reservoir to ventilate the lung. So although I agree that the unidirectional pattern of flow is generated by the geometry of the gas-exchanging part of the lung itself, the air has to come from somewhere, and in exhalation it comes from the air sacs (or the homologous saccular regions of the lung in crocodilians). So the idea that air sacs are unrelated to lung ventilation, which is still implicit in the revised manuscript, is inaccurate.
2. Following directly from that, if air sacs (or homologous saccular regions of the lung) ventilate the gas-exchanging portions of the lung in both crocs and birds, then presumably they have always had that function in archosaurs, and there would seem to be little need to uncover the other functions that drove their evolution. Now, maybe what you mean here is, what selective forces caused bird lungs to become _even more_ heterogeneously partitioned (sensu Perry) than crocodilian lungs. If that’s the case, it would be good to clarify that, and cite Perry’s relevant work here—as with the data from pneumaticity, your paper would benefit from more engagement (i.e., meaningful discussion, not just quick citation) with these relevant and complementary works, not less. Also, as I stated in my original review, “It seems intuitive that the further segregation of the respiratory system into ventilators (air sacs) and gas exchangers (the parabronchial portion of the lung) in birds is related to their high BMR and high activity levels. That hypothesis seems at least as reasonable as the suggestion that the extrapulmonary air sacs of birds and other archosaurs evolved for sound production or locomotion, and it has a long history in the literature (e.g., various papers by Perry), so I am curious why it is not mentioned.”

This is especially true because you already mention that hypothesis in the introduction (lines 67-69): “The archosaur lung may be a key innovation that gave archosaurs a competitive advantage over the synapsids in niches that required _highly aerobic metabolisms_ during the atmospheric hypoxia of the Triassic” (emphasis mine)—if highly aerobic metabolisms were selectively advantageous, the greater efficiency of the fully avian lung/air-sac system over the less heterogeneously partitioned crocodilian respiratory system would seem to be a sufficient driver of that further segregation.

So allow me to strengthen my original recommendation: precisely because that hypothesis has such a long history in the literature, it is imperative that you include it along with the others you mention, and cite the relevant sources.

---

## Round 0.3 · accepted · Accept

Thank you for your detailed attention to the comments from the reviewers, and for the extra work you have undertaken to address suggested changes in the text. In my view, the paper is ready to go. Given the extensive discussion with the reviewers, you might consider posting the reviews and rebuttal related to the manuscript along with the final paper (as PeerJ allows), but of course this is your decision.